# In Silico Prediction of Tetrastatin-Derived Peptide Interactions with αvβ3 and α5β1 Integrins

**DOI:** 10.3390/ph18070940

**Published:** 2025-06-21

**Authors:** Vivien Paturel, Stéphanie Baud, Christophe Schneider, Sylvie Brassart-Pasco

**Affiliations:** 1Université de Reims Champagne-Ardenne, CNRS, MEDYC, F-51100 Reims, France; vivienpaturel@gmail.com (V.P.); christophe.schneider@univ-reims.fr (C.S.); 2Université de Reims Champagne-Ardenne, URCATech, P3M, F-51100 Reims, France

**Keywords:** collagen, integrins, MD simulations, molecular docking, tetrastatin

## Abstract

**Background/Objectives**: Tetrastatin, the globular non collagenous (NC1) domain of the α4 chain of collagen IV, was previously demonstrated to inhibit melanoma progression. We identified the minimal active sequence (QKISRCQVCVKYS: QS-13) that reproduced the anti-tumor effects of whole Tetrastatin and demonstrated its anti-angiogenic activity mediated through αvβ3 and α5β1 binding. As QS-13 peptide was not fully soluble in aqueous solution, we designed new peptides with better water solubility. The present work aimed to investigate the interactions of ten QS-13-derived peptides, exhibiting improved hydro-solubility, with αvβ3 and α5β1 integrins. **Methods**: Using bioinformatics tools such as GROMACS, VMD, and the Autodock4 suite, we investigated the ability of the substituted peptides to bind αvβ3 and α5β1 integrins in silico. **Results**: We demonstrated in silico that all substituted peptides were able to bind both integrins at the RGD-binding site and determined their theoretical binding energy. **Conclusions**: The new soluble peptides should be able to compete with natural integrin ligands such as fibronectin, but also FGF1, FGF2, IGF1, and IGF2. Taken together, these findings suggest that the QS-13-derived peptides are reliable anti-angiogenic and anti-tumor agents.

## 1. Introduction

Integrins are heterodimeric cell surface receptors, composed of non-covalently paired α and β subunits, that bind extracellular matrix (ECM) molecules. In mammals, 18 α-integrin subunits and 8 β-integrin subunits can combine to form as many as 24 unique heterodimeric complexes that lead to various intracellular signals depending on the ligand they bind to. Ligand binding triggers conformational changes in integrins that may interact with other tyrosine kinase receptors [1] and induce different signaling pathways, such as FAK/Src [2], as well as the generation of intracellular forces [3]. Integrins are also biomechanical sensors of ECM stiffness [4].

Alpha v beta 3 integrin binds various extracellular matrix proteins such as fibronectin [5], vitronectin, fibrinogen, von Willebrand factor, osteopontin, and thrombospondin through their RGD motifs, as well as vascular endothelial growth factor receptors (VEGFR) [6,7], fibroblast growth factors 1 and 2 (FGF-1 and 2) [8], and insulin-like growth factor 1 (IGF-1). The primary intracellular downstream signaling mediators of integrins involve focal adhesion kinase (FAK), Src-family protein tyrosine kinases, and integrin-linked kinase (ILK) [9]. This integrin transduces mechanical and biochemical signals to promote cell proliferation, adhesion, spreading, survival, and ECM assembly and remodeling. It is involved in all stages of tumor progression, including tumor initiation, epithelial–mesenchymal transition (EMT), metabolic reprogramming, immune evasion, bone metastasis, tumor neo-angiogenesis, and drug resistance. Initially expressed at low or undetectable levels in most adult epithelial cells, integrin αvβ3 is highly upregulated in many types of cancer. It is induced in so-called “active” endothelial cells during tumor neo-angiogenesis [10].

Alpha 5 beta 1 integrin plays a crucial role in cancer progression by influencing various biological processes. Key ligands of α5β1 integrin include fibronectin, as well as other ECM components such as laminin [11,12,13]. Integrin α5β1 increases the generation of adhesion forces, focal adhesion point assembly, stress fiber formation, and contractile forces. Fibronectin binding to α5β1 integrin activates multiple signaling pathways, including c-Met/FAK/Src-dependent pathways, which are essential for tumor cell invasion and metastasis. It regulates the expression and activity of matrix metalloproteinases (MMPs), particularly MMP-2 and MMP-9. This integrin is also involved in tumor cell survival and proliferation, inducing the activation of kinases such as AKT, mTOR, and ERK1/2. It regulates EGFR-dependent proliferation and Akt-dependent pro-survival in epidermoid carcinoma [14]. It is involved in neo-angiogenesis, favoring endothelial cell proliferation and migration, and regulates angiogenic signals through binding with different partners such as endostatin, VEGFR-1, Angiopoietin-2, and Tie-2. It is highly expressed in various cancer types, including glioblastomas, colon cancer, and breast cancer, where it is often associated with poor treatment response and resistance to apoptosis [15].

Therefore, αvβ3 and α5β1 integrins have been explored as a therapeutic target in cancer treatment for many years. In 1984, the Arg-Gly-Asp (RGD) sequence was discovered in fibronectin as the smallest αvβ3 integrin-binding motif, with an 89nM IC50 [16]. Based on this motif, many peptides have been developed. Cilengitide, a cyclic RGD pentapeptide, and other RGD-based drugs have entered clinical trials. Unfortunately, due to the lack of a curative effect, they failed in phase III trials [17].

Most recently, a 20-mer synthetic peptide that targets both αvβ3 and α5β1 (AXT-107, developed by AsclepiX Therapeutics, Bethesda, MA, USA), derived from the non-collagenous domain of collagen IV, has entered clinical trials for treating retinal vascular diseases [18].

QS-13 (QKISRCQVCVKYS), a bioactive peptide derived from the globular non-collagenous (NC1) domain of the α4 chain of collagen IV, was reported to bind melanoma cells through αvβ3 integrin and to inhibit melanoma progression [19]. It binds close to the RGD-binding site. The interaction between QS-13 and αvβ3 integrin inhibits the FAK/PI3K/Akt pathway, which is essential for tumor cell proliferation and migration. The two cysteine residues form an intrachain disulfide bridge that spontaneously occurs in solution, as highlighted by MALDI-ToF MS analyses. The disulfide bridge appears to be crucial for cell binding [20]. Additionally, QS-13 has been shown to bind endothelial cells through both αvβ3 and α5β1 integrins, and it inhibits angiogenesis both in vitro and in vivo [21].

Unfortunately, QS-13 is insoluble in aqueous solutions and requires prior dissolution in DMSO. To avoid the use of this solvent, we replaced hydrophobic residues (Isoleucine and/or Valine) with more hydrophilic ones (Serine or glycine) and compared the newly designed peptide’s physical properties (stability and hydropathicity index) and 3D structure to the original QS-13 [22]. The pI, charge, and estimated half-life, calculated using the ProtParam tool of Expasy, remained unchanged.

The aim of the present paper is to check the ability of the substituted peptides to bind αvβ3 and α5β1 near the RGD-binding site [23,24,25] in the same way as the QS-13 peptide. To achieve this goal, we employed clustering and docking techniques, as well as energetic and binding analyses, following the same principles as those outlined in previous articles related to QS-13 [19,21].

## 2. Results

To study the interactions between QS-13-derived peptides and αvβ3 and α5β1 integrins, the 3D structures of these two receptors provided by the Protein Data Bank were used. These structures were obtained through X-ray diffraction experiments and corresponded to integrins in a “ligand-free” conformation. Table 1 specifies both integrin PDB IDs and illustrates the respective organization of the α (in blue) and β (in red) subunits within the dimer. Various binding sites were identified on the surface of these integrins. The present study focused on the most distal parts of the extracellular side, corresponding to the RGD-binding domain and, also, to the preferred QS-13 interaction area.

Molecular docking experiments between the QS-13-derived peptides and the integrin molecules shown in Table 1 were carried out in accordance with the strategy used in previous works [19,21].

### 2.1. Substituted Peptide Conformations

Using MALDI-ToF MS, we previously demonstrated that the two cysteine residues of the QS-13 peptide spontaneously formed an intrachain disulfide bridge in solution. Moreover, the substitution of ^222^C and/or ^225^C by an alanine residue abolished peptide binding to the cell and, by the way, also abolished its proliferation, migration, and invasion [20]. To improve QS-13 solubility, hydrophobic amino acids were substituted by hydrophilic ones, and cysteine residues were kept intact. The physicochemical parameters (solubility index, 3D structure) of the newly designed peptides were studied [22]. Based on the results, five of them, with or without a constrained disulfide bridge, were selected for docking studies (Table 2).

To reduce the number of degrees of freedom, conformational sampling of each peptide was achieved through 100 ns molecular dynamics (MD) simulations. The predominant conformations were then extracted through clustering analyses. This preparatory step enabled us to conduct semi-rigid docking, in which the core ligand structures remained rigid, while the side chains were flexible.

Figure 1 depicts the representativeness percentages of the two predominant conformations of each peptide, obtained with a 3 Å cutoff clustering process. Figure 1A shows data associated with peptides devoid of disulfide bonds. Depending on the peptides, the representativeness of the two predominant conformations could differ. QS-13-1 was notably associated with a larger population within Cluster 1, comprising approximately 40% of the total number of conformations explored, in contrast to Cluster 2, which represented less than 10%. A second profile emerged, characterized by a slightly more populated Cluster 1 in comparison to Cluster 2 (QS13-2 and QS13-3 peptides). Finally, in the case of QS13-4 and QS13-5 peptides, Clusters 1 and 2 were almost equally populated. These observations highlighted significant variations in molecular stability from one QS-13-derived peptide to another. In any case, the combined percentages of the two clusters represented over 40% of all conformations explored during a 100 ns molecular dynamics simulation. Considering the peptides containing a disulfide bond and the data shown in Figure 1B, two distinct profiles could be highlighted. The first one revealed that Cluster 1 was much more populated than Cluster 2, as observed in the case of peptides QS-13-1, 2, 3, and 5. Conversely, the QS-13-4 peptide showed a profile in which the population of Cluster 1 was only slightly higher than that of Cluster 2. This highlights the greater stability of peptides with disulfide bridges.

As evidenced by Figure 2, depicting the most representative 3D conformations of QS-13-derived peptides with and without a disulfide bridge, the amino acid substitutions had a minimal impact on the peptide volume. However, as demonstrated in the aforementioned article [22], they significantly impact peptide physicochemical properties, particularly aqueous solubility.

### 2.2. Docking of Substituted Peptides with αvβ3 Integrin

Integrin αvβ3 binds a wide range of ECM molecules containing an Arg-Gly-Asp (RGD) peptide motif, including fibronectin, fibrinogen, von Willebrand factor, vitronectin, and proteolyzed forms of collagen and laminin. This integrin is involved in tumor progression, notably through angiogenesis development. Figure 3 of our study shows the αv integrin subunit in blue and β3 in red. On the right, the integrin dimer is presented in its entirety, with the RGD-dependent binding domain framed in green. Figure 3A–J provide zoomed-in views of the binding site located at the interface between the α and β subunits.

The poses visible in this panel correspond to the lowest binding free energies in Cluster 1 of QS-13-derived peptides with and without constrained disulfide bridges between cysteine residues. It is worth noting that all QS-13 derivatives, whether containing disulfide bridges or not, bind to a similar protein surface. These interaction sites correspond to those of the original QS-13. Consequently, QS-13-derived peptides may compete with ligands binding to αvβ3 in an RGD-dependent manner.

Considering the protein surface covered by the peptide, most of the QS-13-derived peptides are reported to interact more with the β integrin subunit in the binding pocket than with the αv subunit, except for QS-13-3 without a disulfide bridge (Figure 3C) and QS-13-4 with a disulfide bridge (Figure 3I), in the case of which the peptide is mainly located above the αv subunit and therefore reveals a stronger binding to this subunit than to the β3-subunit.

In any case, each of these peptides obstruct the RGD-dependent region of the αvβ3 integrin and may therefore compete with the usual ligands of this domain.

### 2.3. Docking of Substituted Peptides with α5β1 Integrin

Another integrin involved in tumor development and angiogenesis is α5β1. It also binds to fibronectin in an RGD-dependent manner. Consequently, we conducted the same docking experiment on this integrin.

Figure 4 shows α5β1. On the right, we can observe the integrin in its entirety, with a green-bordered area highlighting the QS-13 anchorage site, which also corresponds to an RGD-binding domain.

Figure 4A–J are associated with zoomed-in views of the binding site. As before, the poses visible in this panel correspond to the lowest binding free energy of Cluster 1 for QS-13-derived peptides.

The general behavior of ligands consists of occupying the binding pocket mainly above the β1 subunit. However, two profiles can be identified: in the first one, the entire ligand covers only the surface of the β subunit, as can be observed for QS-13-1 without and with a disulfide bridge (Figure 4A,F), QS-13-2 without a disulfide bridge (Figure 4B), QS-13-3 with a disulfide bridge (Figure 4H), QS-13-4 without and with a disulfide bridge (Figure 4D,I), and QS-13-5 with a disulfide bridge (Figure 4J). The second binding profile is characterized by the fact that the ligand also covers a small portion of the α subunit, as observed with peptide QS-13-2 with a disulfide bridge (Figure 4G), QS-13-3 without a disulfide bridge (Figure 4C), and QS-13-5 without a disulfide bridge (Figure 4E).

As with αvβ3 integrin, all QS-13 derivatives, with disulfide bridges or not, bind to the same area and may compete with RGD-dependent ligands for α5β1 binding.

We performed an RMSD analysis on the predominant conformations of clusters obtained from the top 150 docking poses to evaluate the stability and variability of the structures based on the presence or absence of disulfide bridges and the peptide sequence on α5β1 and αvβ3 integrins (Appendix A). This analysis measured the similarities and differences between the poses. Overall, the majority of conformations exhibited RMSD values ranging between 10 Å and 15 Å. However, an exceptional case stood out, notably in Appendix A), where the QS-13-5 pose without a disulfide bridge on α5β1 integrin shows higher RMSD values, with one group of conformations around 10 Å and another group reaching 18 Å, indicating significant structural flexibility in this specific case.

We superimposed the two best docking poses on the co-crystallized integrins to verify, based on experimental data, that the peptides bound to the same site as the co-crystallized RGD peptides. Appendix A shows the superposition of the two best docking poses from the two predominant clusters of peptides (with and without disulfide bridges) on the co-crystallized structures of integrins with RGD peptides. The dockings were performed on the structures of α5β1 integrin (PDB ID: 3VI3) and αvβ3 integrin (PDB ID: 4G1M), but the visualization was performed on co-crystallized structures with RGD peptides. In Appendix A, the docking was performed on the α5β1 integrin, where Appendix A represents peptides without disulfide bridges and Appendix A those with disulfide bridges. Similarly, Appendix A show the docking results on the αvβ3 integrin, where C corresponds to peptides without disulfide bridges and D to peptides with disulfide bridges. It could be observed that all peptides derived from QS-13, whether with or without disulfide bridges and on both integrins, occupy the same binding site as the co-crystallized RGD peptide.

### 2.4. Binding Energy Analysis

Alongside this structural study, energetic data probing the strength of the interaction between QS-13-derived peptides and integrins were also collected, generating the histograms shown in Figure 5. A visual representation of the binding free energies (expressed in kcal/mol) associated with the best peptide poses of each cluster on the two integrins surfaces was thus obtained. It is worth noting that lower energies signify stronger interactions, as the protein/ligand complex was then more energetically stable than the two separate molecular actors.

For the reference peptide, QS-13, which was not included in Figure 5, the binding free energies were as follows:-For αvβ3: −5.70 kcal/mol and −7.40 kcal/mol, respectively, for conformations 1 and 2.-For α5β1: −5.71 kcal/mol and −7.25 kcal/mol, respectively, for conformations 1 and 2 [21].

Figure 5A illustrates the binding free energies of peptides derived from QS-13 without a disulfide bridge on integrin αvβ3. These binding energies were around −6.00 kcal/mol, and it was interesting to observe that no peptide stood out from the others on this protein target.

Figure 5B explores the binding energies of peptides derived from QS-13 with a disulfide bridge on the αvβ3 integrin. The presence of the disulfide bridge generally led to a reduction in binding energy, thereby strengthening the anchoring. The energy of QS-13-1 did not change significantly, while the binding energies of the other four peptides were modified. For instance, QS-13-2 exhibited a binding free energy of −9.98 kcal/mol, representing a reduction of approximately −4.00 kcal/mol, while the other peptides displayed binding free energies ranging between −8.50 kcal/mol and −6.43 kcal/mol, which indicated a decrease of 1.00 to 2.00 kcal/mol. Regarding the possible distinction between the two most representative peptide conformations, very similar binding free energies with minor variations ranging from 0 to 1.00 kcal/mol could be observed, except for the QS-13-2 peptide with a disulfide bridge, for which a notable difference of 3.50 kcal/mol could be highlighted.

Figure 5C represents the binding energy of peptides derived from QS-13 without a disulfide bridge on the α5β1 integrin. These energies varied, ranging from approximately −7.52 kcal/mol for the second conformation of QS-13-2 to −2.74 kcal/mol for the second conformation of QS-13-4. The profiles were diverse, sometimes showing a dominant conformation with a better binding free energy than with that of the second conformation, as is the case for QS-13-4. In other instances, the binding free energy was more favorable in the case of the second peptide conformation. Finally, overall, the binding free energies of the derived peptides were slightly higher, and therefore less favorable, than those of the original QS-13. Figure 5D is dedicated to peptides that have a disulfide bridge between cysteine residues and interact with the α5β1 integrin. The binding energies of these derivatives of the QS-13 peptide varied from −7.27 kcal/mol for the lowest to −3.76 kcal/mol for the highest. Most peptides had binding energies between −7.50 kcal/mol and −6.00 kcal/mol. In general, the binding energies on this integrin were lower than those of the original QS-13, thus strengthening the peptide/protein interaction.

In addition to examining the binding free energy of the best conformations, for each integrin/QS-13-derived complex, the binding energies corresponding to the three best conformations were also collected and are presented in Table 3. The superimposition of the corresponding poses is illustrated in Appendix A (complexes with αvβ3 integrin) and Appendix A (complexes with α5β1 integrin). For almost all complexes, the difference in binding free energy between the best solution and the third solution is less than 1 kcal/mol; moreover, superimposing these three poses often highlights a preferred orientation. These observations allow us to continue the study by considering only the best pose for each complex.

### 2.5. Determination of the Amino Acids Involved in the Interactions

In addition to the analysis of binding free energies, the coordinates of the ligand poses made it possible to characterize the amino acids involved in the interactions to assess potential impacts on cellular behavior, particularly in terms of potential competition with other ligands.

A comparative analysis of the most representative conformations with and without disulfide bridge revealed interesting binding patterns. Two types of interactions were identified at the binding site: hydrophobic contacts and hydrogen bonds. No π-π interactions were observed, and T-π interactions and cation-π interactions were observed only once each among the 40 complexes studied.

The interaction between QS-13-derived peptides without disulfide bridges and integrin αvβ3 involved recurrent amino acid residues from the integrin (Table 4 and Table 5 and Appendix A). Thus, Ala215 and Tyr178 of the αv subunit and residue Asn313 of the β3 subunit interacted with five, four, and four poses of QS-13-derived peptides, respectively. These results suggested a high affinity of these peptides for these specific integrin sites.

Regarding interactions between disulfide bridge-containing QS-13-derived peptides and integrin avβ3, recurrent binding sites were also observed (Table 6 and Table 7 and Appendix A). Residues Tyr178 and Ala215 from the αv subunit and residue Tyr122 from the β3 subunit interacted with three, six, and three poses of QS-13-derived peptides, respectively.

To sum up, these results indicate that QS-13-derived peptides, whether or not they possess a disulfide bridge, had a high affinity for specific αvβ3 integrin residues, such as Ala215 and Tyr178 from the αv subunit. Other residues, such as Tyr122, Met180, Arg214, Asn313, and Met335 from the β3 subunit were highlighted due to having an interaction with different peptide poses which did or did not contain a disulfide bridge. An overlap of the binding zone of QS-13-derived peptides with that of the RGD-binding domain was observed. Furthermore, many of the amino acids identified in this study, such as Met180, Met335, Ser123, and Tyr122, had already been identified as interacting with the original QS-13. In any case, as for the few peptides not interacting with these domains, they are located in the RGD-dependent zone and thus close to the original QS-13 binding site [19].

Then, the binding site between the QS-13-derived peptides and the α5β1 integrin was also characterized.

Regarding the interactions established by QS-13-derived peptides without disulfide bridge, several integrin residues have been identified as preferred binding sites (Table 8 and Table 9 and Appendix A). Thus, residues Trp157, Phe187, Ile225, and Asp227 of the α5 subunit and also residues Ser134, Tyr133, and Lys182 of the β1 subunit established interactions with four, five, three, three, three, and three poses of QS-13-derived peptides, respectively.

Regarding the interactions between QS-13-derived peptides with a disulfide bridge and the α5β1 integrin, residues Phe187, Ser 224, Asp227, and Trp157 of the α5 subunit and residues Ser134 and Lys182 of the β1 subunit played key roles in the interactions with 10, 5, 4, 3, 4, and 3 poses of the peptides derived from QS-13, respectively, and are thus highlighted (Table 10 and Table 11 and Appendix A).

Residues Phe187, Trp157, Ser224, and Asp227 of the α5 subunit (with 15, 7, 7, and 7 interactions, respectively) and Ser134, Lys182, and Tyr133 of the β1 subunit (with 7, 6, and 5 interactions with poses of the derived peptides, respectively) were involved in binding with QS-13-derived peptides with or without a disulfide bridge. This highlights their important role in binding. In summary, our results point out that QS-13-derived peptides interact with specific residues of α5β1 integrin, whether or not they possess a disulfide bridge. Asp227 of the α5 subunit and Ser134, Asp137, Glu189, Glu221, and Glu320 of the β1 subunit have previously been shown to interact with the native QS-13 peptide. Thus, the proposed modifications within the peptide sequence did not modify the interaction area, which allows us to assume that peptides derived from QS-13 are likely to conserve the biological effects of the original QS-13 [21].

## 3. Discussion

Integrins not only mediate adhesive interactions between cells and their environment but are also involved in outside–in signaling and control many aspects of cell behavior, including survival, proliferation, metabolism, and differentiation, as well as cell shape and motility, and are involved in cancer progression. αvβ3 and α5β1 integrins are not only expressed on tumor cells; they are also induced on endothelial cells during the process of angiogenesis [26]. These integrins have been reported as being targets for anti-angiogenic therapies in cancer, and investigations in clinical trials proved disappointing. Humanized monoclonal antibodies such as Vitaxin, Abegrin, Volociximab, RGD-derived peptides, and RGD-mimetic molecules have been investigated in clinical trials. The results in cancer treatment turned out to be dissatisfying [27].

We previously identified a 13 amino-acid peptide from the α4(IV) collagen chain, located in the NC1 domain, which binds to both integrins in close vicinity to the RGD-binding site, reducing tumor and endothelial cell migration and thus, consequently, tumor progression. Unfortunately, this peptide was not soluble in the aqueous medium and required prior dissolution in DMSO. In a previous paper, hydrophobic amino acid substitutions were proposed to increase peptide solubility, without, however, modifying its secondary structure too drastically [22].

In the present article, clustering experiments performed on 100 ns molecular dynamics simulations suggested a better stability of the peptides with constrained disulfide-bridge, since the overall representativeness of the first two conformations was increased.

In docking experiments with substituted peptides on αvβ3 and α5β1 integrins, the binding free energies evaluated were of the same order of magnitude as those obtained with the original QS-13 peptide. Some substitutions and the introduction of a constrained disulfide bridge even improved these energies.

Close examination of the interaction sites revealed the involvement of specific integrin residues.

Interactions of the substituted QS-13 peptide with integrin αvβ3 are characterized by the frequent involvement of the following amino acids: Tyr178 (αv subunit), Tyr122, Ser123, Met180, and Met335 (β3 subunit). These residues were also involved in original QS-13 binding. Amino acid substitutions did not seem to alter the interaction sites on integrin αvβ3 to any great extent. However, one aspect that is not considered when setting up docking experiments is the fact that the target protein is considered rigid. Integrins are known to be flexible and can adopt different conformations upon ligand binding. So far, the in silico results obtained with the canonical QS-13 peptide have been consistent with the available experimental data [19,21]. However, in future research, it will be interesting to study the intrinsic flexibility of the ligand-free protein prior to docking experiments and to test the robustness of the formed complex ligand-bound structures through MD extensive simulations.

Integrin αvβ3 binds various ECM proteins, such as fibronectin, vitronectin, von Willebrand factor, fibrinogen, osteopontin, and thrombospondin, through their RGD motif, but also Vascular Endothelial Growth Factor (VEGF) receptors, Fibroblast Growth Factor-1 and 2 (FGF-1 and 2), and Insulin Growth Factor 1 (IGF-1), promoting cancer cell invasiveness [14].

Interactions of the substituted QS-13 peptide with integrin αvβ3 were characterized by the frequent involvement of the following amino acids: Tyr178 (αv subunit), Tyr122, Ser123, Met180, and Met335 (β3 subunit). Tyr122, Ser123, Met180, and Met335 were also involved in original QS-13 binding. Amino acid substitutions did not seem to alter the interaction sites on integrin αvβ3 to any great extent.

The Tyr178 residue of the αv subunit appeared crucial for αvβ3 adhesion to fibrinogen, as previously reported. Substitution of the Tyrosine residue at position 178 with an Alanine residue (Tyr178Ala) was previously reported to significantly impair cell adhesion to immobilized fibrinogen [28,29]. The Tyr122 residue of the β3 subunit played a crucial role in the interaction with fibronectin as revealed by crystallographic analysis [29]. Moreover, residue Tyr178 of the αv subunit and residues Tyr122, Ser123, and Met180 of the β3 subunit were involved in FGF1 binding [30]. These same residues, as well as the Met335 residue of the β3 subunit, were also involved in FGF2 binding [8]. The Tyr178 residue of the αv subunit and the Ser223 residue of the β3 subunit were also reported to be involved in the IGF2 interaction. Regarding these data, QS-13-derived peptides could compete with macromolecules such as fibrinogen for αvβ3 integrin binding, but also with growth factors, and thus inhibit the pro-tumoral transduction pathways induced by these molecules.

Regarding the α5β1 integrin, residues Trp157, Phe187, Ser224, and Asp227 from the α5 subunit and residues Tyr133, Ser134, and Lys182 from the β1 subunit seemed to be involved in the binding of QS-13-derived peptide. Residues Asp227 (α5 subunit), Ser134, Ser189, Ser227, and Glu320 (β1 subunit) were previously reported as interaction sites with the original QS-13 peptide [21]. The topology of the binding area of the newly designed peptides appeared very similar to that of the original QS-13 binding area and therefore to the RGD-binding area. The RGD-binding domain of integrin α5β1 is crucial in various biological and pathological processes, as described in the Introduction section. It plays an essential role in the regulation of angiogenesis and tumor malignancy [14].

A previous structural model of the human α5β1 integrin confirmed the importance of residues Trp157 and Ala158 in the antagonistic binding of this integrin [11]. The Trp157 residue is notably involved in a hydrophobic interaction with the tryptophan residue (W) in RGDGW-type sequences; this interaction determines the specificity of the integrin α5β1 for RGDGW-containing peptides [31]. The Ser156 and Trp157 residues are part of the receptor binding pocket and blocking these two amino acids prevents α5β1-mediated cell adhesion [32]. Site-directed mutagenesis demonstrated that in addition to the previously identified residue Asp130, residues Ser132, Asn224, Asp226, Glu229, Asp233, Asp267, and Asp295 are also required for adhesion to fibronectin (FN). Indeed, the mutation of these residues prevented binding to immobilized FN 110K fragments and to soluble FN [12]. Fibronectin made a strong van der Waals contact with the α1 helix backbone adjacent to Ser134 and packed against Tyr133 in the β1 domain. Close to the RGD loop, FN10 formed a salt bridge with the Glu320 residue of the β1 subunit [13]. Regarding these data, QS-13-derived peptides could compete with fibrinogen for α5β1 integrin binding and prevent angiogenesis and tumor progression.

Additionally, the RMSD distribution for all the best poses obtained from the 150 docking runs is predominantly in the range of 10 to 13 Å, which we consider reasonable given the length of the peptide (13 amino acids). To improve the predictive power of our models, we compared our docking results to experimental data, where the superposition of all the peptide poses onto co-crystallized integrin structures showed a significant overlap with the RGD-binding region across both integrins (αvβ3 and α5β1).

Integrin αvβ3 and α5β1 are also involved in a variety of pathological process other than tumor progression, including cardiac hypertrophy, atherosclerosis [33], psoriasis [34], rheumatoid arthritis [35], osteoporosis [36], diabetic nephropathy [37], retinopathy [38], and macular edema. For example, THR-687, a small molecule, and AG-73305, a fusion protein, were used in the treatment in diabetic macular edema to target these integrins. Additionally, PLN-74809, which specifically targets αvβ3, has been used in the treatment of keloids. A 20-mer synthetic peptide that targets both αvβ3 and α5β1, AXT-107, derived from the non-collagenous domain of collagen IV, has now entered clinical trials for treating retinal vascular diseases [39,40].

Our results suggest that the new peptides derived from QS13 may prevent the binding of the usual molecular partners of αvβ3 and α5β1 integrins and, consequently, prevent tumor development and other pathologies listed above. Although AutoDock4 presents some limitations, previous studies from our laboratory have shown consistent results between in silico and in vitro experiments [19,21,41]. Moreover, the results obtained with AutoDock4 were compared with those obtained with AutoDock CrankPep [42], an approach used to dock flexible peptides into rigid receptors, and the results were similar.

Further studies will be needed to confirm this hypothesis. Plasmon resonance studies will be performed to verify that the amino acid substitutions will not alter their interaction with the integrins. Then, the most active peptide will be tested using in vitro and in vivo models.

## 4. Materials and Methods

### 4.1. Peptide Modeling and Target Preparation

Five substituted peptides (presented in Table 2), with or without a constrained intrachain disulfide bridge, were selected according to their physicochemical parameters (solubility index, 3D structure) as previously published [22]. Gromacs software (version 2020.1) [43,44] was used to perform peptide molecular dynamics simulations.

The five modified peptides, along with the original QS-13, were modeled in an extended conformation using the Molefacture extension of the Visual Molecular Dynamics (VMD) software for LINUXAMD64, version 1.9.4a55 accessed on 18 October 2021 [45]. The peptides were capped at the N-terminus (NT2) and C-terminus (ACE) using the AutoPSF extension in VMD. Subsequent molecular dynamics (MD) simulations were conducted using GROMACS (version 2020.1) [43,44] with the OPLSAA (Optimized Potentials for Liquid Simulations All-Atom) force field [45], which was applied to model molecular interactions accurately.

The initial structures for the MD simulations were selected by monitoring the distance between the sulfur atoms of the cysteine residues in peptides without a disulfide bond (DB) throughout the simulation trajectory. The structures with the smallest sulfur atom distance were then chosen and used to generate peptides with a disulfide bond. Using these initial conformations, MD simulations were performed following the previously described protocol.

Each peptide was solvated with counterions, ensuring neutral charge conditions for the MD simulations. Water molecules were represented using the TIP3P model [46], creating an environment suitable for studying molecular interactions under physiological conditions. For solvation in DMSO, parameters from the OPLSAA force field [47] were used to construct the DMSO topology.

All structures underwent energy minimization to obtain stable starting conformations. The steepest descent algorithm was applied for a maximum of 50,000 steps, with an energy tolerance of 1000.0 kJ/mol/nm and a maximum displacement step of 0.01 nm.

The systems were equilibrated in two phases. First, an NVT (constant Number of particles, Volume, and Temperature) ensemble was simulated for 160 ps. Initial velocities were assigned based on the Maxwell distribution at 310 K. A time step of 0.002 ps (2 fs) was employed using the leap-frog integration algorithm, with hydrogen bonds constrained via the LINCS algorithm [48]. The V-rescale thermostat [49] was applied for temperature coupling with a time constant of 0.1 ps, maintaining the system at a reference temperature of 310 K for both protein and nonprotein components. Nonbonded interactions were handled using a Coulomb cutoff of 1.0 nm and a van der Waals cutoff of 1.0 nm, while long-range electrostatics were calculated using the Particle Mesh Ewald (PME) [50] method. Following the NVT simulation, an NPT (constant Number of particles, Pressure, and Temperature) ensemble was simulated for 160 ps to equilibrate the system at 1.0 bar pressure, using the Parrinello–Rahman algorithm [51] with a time constant of 2.0 ps. Other integration and interaction parameters remained the same as those in the NVT phase.

Finally, NPT production runs were carried out for 100 ns. Given the size of the peptides, the comparative nature of the study, and prior experience with matrikine studies [19,52,53,54], this sampling time was deemed sufficient. Moreover, the simulation results for canonical peptides aligned with previously published data [19]. Throughout both the equilibration and production phases, periodic boundary conditions were applied in all three dimensions, and the system coordinates, velocities, and energies were saved every 1.0 ps.

Table 1 shows the two integrin structures that were considered to set up the docking experiments. The Protein Data Bank (PDB) files of the receptors were retrieved from the RCSB-PDB server, and both sets of coordinates corresponded to X-ray diffraction experiments. PDB-IDs 4G1M [55] and 3VI3 [56] were selected to characterize putative interactions with integrin αvβ3 and integrin α5β1, respectively.

The α and β subunits of the crystallographic structures were prepared for the docking experiment with the python-based MGLTools software, mgltools_x86_64Linux2_1.5.6 accessed on 29 October 2022 [57] using the Autodock4 suite [58,59]. Preparation steps included removing water molecules, adding hydrogen atoms, checking for missing atoms, and repairing if necessary. Assignment of Kollmann charges, as well as verification and adjustment of charges, were carried out where necessary. Docking parameters were carefully adjusted to reproduce the experimental results and then investigate potential interactions between QS-13-derived peptides and the αvβ3 or α5β1 receptors.

### 4.2. Docking Experiments

As stated previously, the ligands under investigation were derived from QS-13. They were prepared as semi-flexible molecules: the most representative conformations were extracted from molecular dynamics simulations by clustering analysis [21] and, while the backbone remained fixed, the side chains allowed them to be flexible. To comply with the limit of 32 degrees of freedom, in addition to the backbone rotations, rotations around the distal carbons of lysine residues were also disabled.

Docking experiments between QS-13-derived peptides and integrins were performed using Autodock4 software [39]. Potential binding sites, as reported previously [19,21], were targeted using a large cubic box (including the RGD targeted surface) with a volume equal to 47.25 Å × 47.25 Å × 47.25 Å and a spacing equal to 0.375 Å along each of the three directions. The coordinates of the center of this box were (−18.00, −21.00, −12.00) in the case of α5β1 and (15.21, 41.58, 43.37) in the case of αvβ3.

The Lamarckian genetic algorithm was used to perform 150 independent docking experiments with each cluster. The population size was set at 250 individuals, while the total number of evaluations and the maximum number of generations were 2,500,000 and 27,000, respectively.

### 4.3. Binding Free Energy and Network of Interaction

Upon completion of each docking simulation, binding free energies between ligands and receptors were analyzed along with the network of specific interactions within the RGD-dependent domains.

Ligplot+ [60] and Binana [61] was used to assess the interaction network between each QS-13 derivative and the two integrins.

### 4.4. Docking Pose Representations

The best poses extracted from docking simulations were selected and were showcased using VMD [14]. On the one hand, integrins were visualized using the quicksurf representation. The ligands, on the other hand, were depicted in licorice style, excluding hydrogen atoms for clarity. Such representation highlighted the spatial interactions between the ligands and integrins, providing a visual insight into binding sites and preferred conformations.

The best docking pose from the two predominant clusters of different peptides (both with and without disulfide bridges) was also superimposed onto the crystallized structures of integrins 3VI3 (α5β1) and 4G1M (αvβ3). These superimposed poses were then aligned with 3VI4 (α5β1) [62] and 1L5G (αvβ3) [63] structures, which were co-crystallized with RGD peptides. This superposition allowed comparison between the predicted binding conformations of the docked peptides and the known crystal structures of the RGD-bound integrins.

### 4.5. RMSD Calculations

The docking log files generated from the top 150 docking poses were processed using a custom-made Python3.12.3 script to calculate the Root Mean Square Deviation (RMSD) between these poses. The atomic coordinates of the docking poses were extracted, and a matrix of RMSD values was computed for each pair of poses. Since the matrix was symmetrical (i.e., the RMSD between pose 1 and pose 2 was identical to the RMSD between pose 2 and pose 1), only the upper triangular part of the matrix was used, excluding the diagonal, which contained trivial comparisons of each pose with itself (RMSD = 0). The resulting RMSD values were expressed in Ångströms (Å), and a distribution of these values was created using intervals of 0.5 Å. This distribution provided an insight into the variability and similarity of the docking poses, helping to evaluate the quality and diversity of the top-ranked docking conformations. The GitHub repository with the script is available at the following link: https://github.com/vivienpaturel/-RMSD-.dlg-Autodock4- accessed on 1 September 2024.

## 5. Conclusions

This study allowed us to determine the potential binding sites and binding energy of newly designed peptides on αvβ3 and α5β1, to compare them with original QS-13 peptide, and to check their ability to compete on the RGD-binding site with fibronectin, but also with other integrin ligands, such as FGF1, FGF2, IGF1, and IGF2, which are involved in many physio-pathological processes.

In conclusion, we designed new peptides that are fully soluble in aqueous solution but retain their theoretical ability to interact with αvβ3 and α5β1 integrins. This study will pave the way to in vitro and in vivo studies for future therapeutic applications.

## Figures and Tables

**Figure 1 pharmaceuticals-18-00940-f001:**
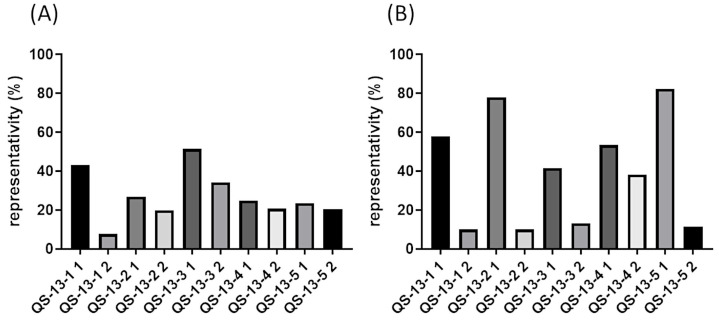
Representativeness of the two predominant conformations of QS-13-derived peptides observed by clustering (3 Å cutoff) in a 100 ns molecular dynamics simulation. (**A**): Peptides without a disulfide bond between cysteine residues; (**B**): peptides with a disulfide bond between cysteine residues.

**Figure 2 pharmaceuticals-18-00940-f002:**
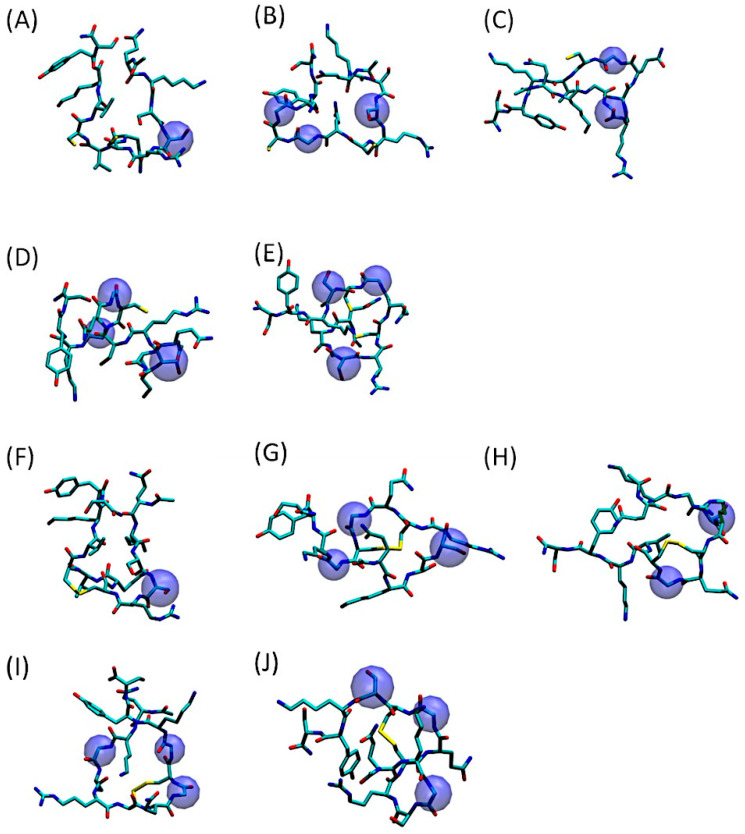
Three-dimensional representation of the predominant conformation of QS-13-derived peptides. The amino acids substituted in the original QS-13 are highlighted in blue spheres. Molecular dynamics simulations were processed using the gromos clustering algorithm for peptides without (**A**–**E**) or with a disulfide bond (**F**–**J**). (**A**,**F**): QS-13-1; (**B**,**G**): QS-13-2; (**C**,**H**): QS-13-3; (**D**,**I**): QS-13-4 (**E**,**J**): QS-13-5.

**Figure 3 pharmaceuticals-18-00940-f003:**
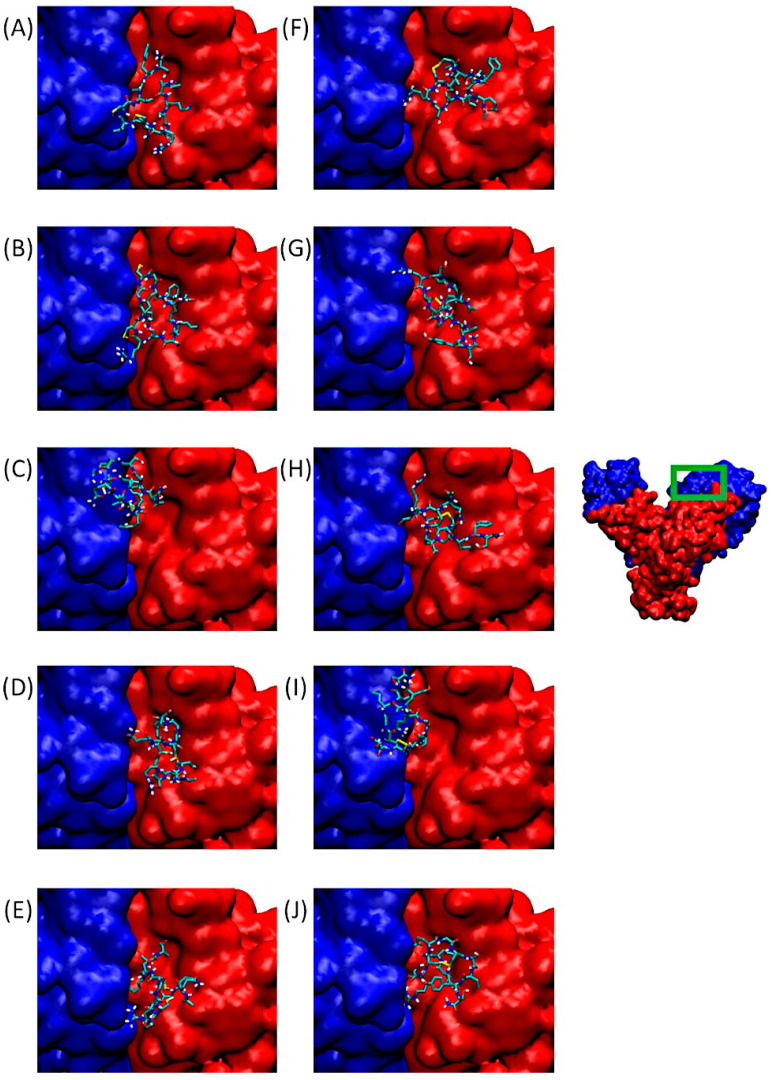
Predominant conformations of peptides derived from QS-13 docked onto the αv (blue surface) β3 (red surface) integrin. The area framed in green on the integrin has been zoomed for better visibility. (**A**–**E**): Peptides without disulfide bridge between cysteine residues; (**F**–**J**): peptides with a disulfide bridge. (**A**,**F**): QS-13-1, (**B**,**G**): QS-13-2, (**C**,**H**): QS-13-3, (**D**,**I**): QS-13-4, (**E**,**J**): QS-13-5.

**Figure 4 pharmaceuticals-18-00940-f004:**
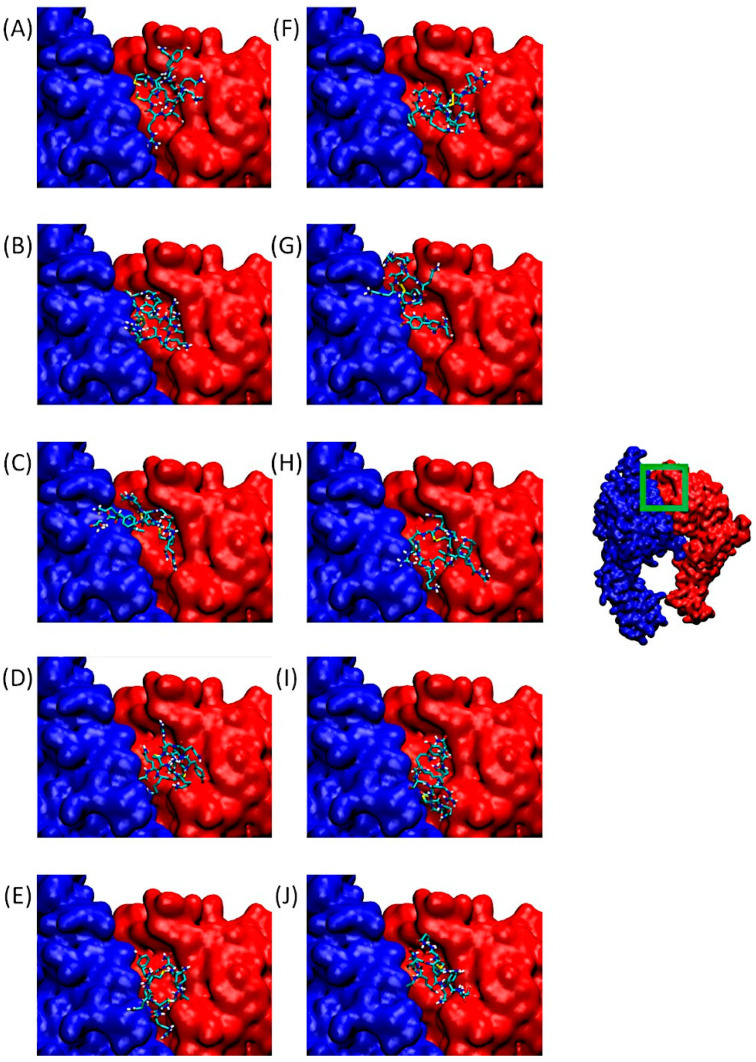
Predominant conformations of peptides derived from QS-13 docked onto the integrin α5 (blue surface) β1 (red surface). The area framed in green on the integrin has been zoomed for better visibility. Figures (**A**–**E**) correspond to peptides without disulfide bridges between cysteine residues, while figures (**F**–**J**) are associated with peptides featuring a disulfide bridge. (**A**,**F**): QS-13-1, (**B**,**G**): QS-13-2, (**C**,**H**): QS-13-3, (**D**,**I**): QS-13-4, (**E**,**J**): QS-13-5.

**Figure 5 pharmaceuticals-18-00940-f005:**
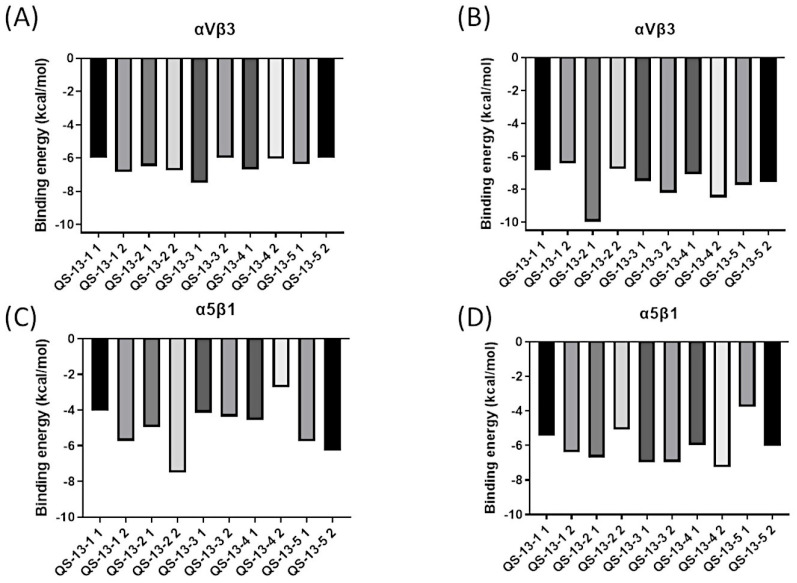
Binding free energy (in kcal/mol) for QS-13-derived peptides. Binding energies for QS-13-derived peptides on the αvβ3 integrin without disulfide bridges between cysteine residues (**A**), with disulfide bridges between cysteine residues (**B**), on the α5β1 integrin without disulfide bridges between cysteine residues (**C**), and with disulfide bridges between cysteine residues (**D**).

**Table 1 pharmaceuticals-18-00940-t001:** Integrin targets used in the molecular docking experiments.

PDB ID	Source	Name of the Protein and Experimental Technics	3D Structure
4G1M	Homo sapiens	αvβ3 integrin, X-ray diffraction (2.90 Å resolution)	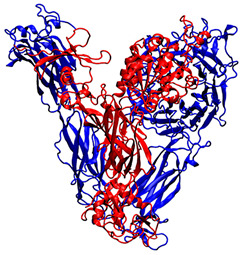
3VI3	Homo sapiens	α5β1 integrin, X-ray diffraction (2.90 Å resolution)	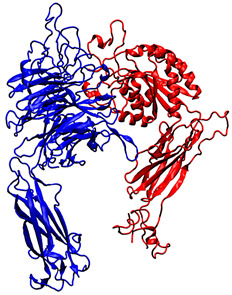

For each PDB identifier used, the experimental technique for structural determination, the associated resolution, and the organism of origin of each protein are specified. The overall organization of the dimers is displayed with new cartoon representation of αvβ3 and α5β1integrins (α subunits are colored in blue and β subunits in red). These integrins are visualized from the extracellular side upwards and the transmembrane side downwards.

**Table 2 pharmaceuticals-18-00940-t002:** Names of the water-soluble peptides derived from QS-13 and the corresponding sequences.

**Peptide**	**Sequence**
QS-13	QKISRCQVCVKYS
QS-13-1	QK**S**SRCQVCVKYS
QS-13-2	QK**S**SRCQ**G**C**G**KYS
QS-13-3	QK**G**SRCQ**G**CVKYS
QS-13-4	QK**G**SRCQ**G**C**G**KYS
QS-13-5	QK**G**SRCQ**G**C**S**KYS

Bold characters indicate mutated residues.

**Table 3 pharmaceuticals-18-00940-t003:** Binding free energy (in kcal/mol) of the three best poses of QS-13-derived peptides in the different complexes investigated.

**Most Representative** **Conformation**	**Integrin α5β1**		**Integrin αvβ3**	
	**With Disulfide** **Bridge**	**Without Disulfide** **Bridge**	**With Disulfide** **Bridge**	**Without Disulfide** **Bridge**
QS-13-1	−5.44	−4.05	−6.84	−5.97
	−5.25	−3.56	−6.67	−5.90
	−4.71	−2.89	−6.58	−5.45
QS-13-2	−6.70	−4.97	−9.98	−6.50
	−6.58	−4.92	−9.65	−6.19
	−6.12	−4.65	−9.48	−5.97
QS-13-3	−6.98	−4.16	−7.49	−7.52
	−6.40	−2.37	−7.49	−5.72
	−6.38	−2.17	−7.21	−5.51
QS-13-4	−5.99	−4.57	−7.11	−6.71
	−5.42	−4.48	−6.10	−5.95
	−4.93	−4.08	−5.92	−5.75
QS-13-5	−3.76	−5.76	−7.74	−5.79
	−3.72	−5.12	−6.40	−5.64
	−3.60	−4.12	−6.27	−5.62
**Second Most Representative** **Conformation**	**Integrin α5β1**		**Integrin αVβ3**	
	**With Disulfide** **Bridge**	**Without Disulfide** **Bridge**	**With Disulfide** **Bridge**	**Without Disulfide** **Bridge**
QS-13-1	−6.39	−5.73	−6.43	−6.86
	−4.19	−5.17	−6.10	−6.36
	−3.66	−3.88	−5.82	−5.78
QS-13-2	−5.09	−7.52	−6.75	−6.77
	−5.03	−4.31	−6.38	−6.63
	−4.92	−4.00	−6.30	−5.90
QS-13-3	−6.96	−4.38	−8.22	−6.00
	−5.77	−4.02	−7.95	−5.89
	−5.58	−3.79	−7.78	−5.87
QS-13-4	−7.27	−2.74	−8.50	−6.03
	−6.78	−2.62	−8.15	−5.18
	−6.55	−2.45	−7.78	−4.86
QS-13-5	−6.01	−6.26	−7.56	−7.84
	−6.01	−6.22	−7.39	−7.71
	−5.71	−5.46	−7.35	−7.09

**Table 4 pharmaceuticals-18-00940-t004:** List of the interactions identified between the most representative conformation of QS-13-derived peptides without a disulfide bridge and αvβ3 integrin.

Peptides Without Disulfide Bond—Most Representative Conformation	αv	β3
QS-13-1	**Hydrophobic contact**	
	Cys9—Ala215	
	**Hydrogen bond**	
	Val8 (C=O)—Ala215 (N)	
	Val8 (C=O)—Ile216 (N)	
QS-13-2	**Hydrophobic contact**	
	Arg5—Ile216	Cys9—Arg214
	Cys6—Ala215	
	Gly8—Tyr178	
	Cys9—Ty178	
	**Hydrogen bond**	
		Cys9 (C=0)—Arg214 (N sidechain)
QS-13-3	**Hydrophobic contact**	
	Val10—Trp179	Ser13—Ala218
	Lys11—Tyr178	
	Lys11—Trp179	
	Tyr12—Tyr178	
	Tyr12—Asp219	
	**Hydrogen bond**	
		Ser13 (C=O)—Ala 218 (N)
QS-13-4	**Hydrophobic contact**	
		Nter—Met335
		Gly3—Asp251
		Ser4—Asn313
		Lys11—Tyr122
		Tyr12—Arg214
		Cter—Met180
	**Hydrogen bond**	
		Lys11 (C=O)—Arg214 (N sidechain)
QS-13-5	**Hydrophobic contact**	
		Nter—Met335
		Gly3—Asn313
		Ser4—Asn313
		Arg5—Asn313
		Ser13—Arg214
		Ser13—Asn215
	**Hydrogen bond**	
		Cter (N)—Asn215 (O sidechain)
		Cter (N)—Arg216 (O sidechain)

The nature of the interaction is classified as hydrophobic contact (the residues involved in the contact are specified) or hydrogen bonding (the residues and atoms involved in the contact are specified).

**Table 5 pharmaceuticals-18-00940-t005:** List of the interactions identified between the second most representative conformation of QS-13-derived peptides without a disulfide bridge and αvβ3 integrin.

Peptides Without Disulfide Bond—Second Most Representative Conformation	αv	β3
QS-13-1	**Hydrophobic contact**	
	Arg5—Ala218	
	Cys6—Ala215	
	Gln7—Tyr178	
	Val8—Tyr178	
	Tyr12—Asp148	
	**Hydrogen bond**	
		Cter (N)—Arg216 (N sidechain)
QS-13-2	**Hydrophobic contact**	
	Nter—Tyr178	
	Gln1—Ala215	
	Lys2—Tyr178	
	**Hydrogen bond**	
	Gln1 (N)—Asp218 (O sidechain)	
QS-13-3	**Hydrophobic contact**	
		Nter—Ser123
		Nter—Asp251
		Ser4—Ser123
		Arg5—Ser123
		Val10—Met180
		Ser13—Asn215
		Ser13—Arg216
		Ser13—Asp217
		Ser13—Ala218
	**Hydrogen bond**	
		Gln1 (N)—Asp251 (C=O)
QS-13-4	**Hydrophobic contact**	
	Nter—Ala215	
	Nter—Asp218	
QS-13-5	**Hydrophobic contact**	
		Nter—Met335
		Gln1—Asp251
		Ser13—Asn313
		Cter—Asn313
	**Hydrogen bond**	
		Nter (C=O)—Thr311 (O sidechain)

The nature of the interaction is classified as hydrophobic contact (the residues involved in the contact are specified) or hydrogen bonding (the residues and atoms involved in the contact are specified).

**Table 6 pharmaceuticals-18-00940-t006:** List of the interactions identified between the most representative conformation of QS-13-derived peptides with a disulfide bridge and αvβ3 integrin.

Peptides with Disulfide Bond—Most Representative Conformation	αv	β3
QS-13-1	**Hydrophobic contact**	
		Gln7—Ser123
		Val8—Tyr122
		Cys9—Met180
QS-13-2	**Hydrophobic contact**	
	Arg5—Tyr178	
QS-13-3	**Hydrophobic contact**	
	Ser4—Tyr178	
	Arg5—Tyr178	
QS-13-4	**Hydrophobic contact**	
	Cys6—Tyr178	
	Gln7—Ala215	
	Lys11—Asp150	
	Tyr12—Asp148	
	**Hydrogen bond**	
	Ser13 (C=O)—Lys119 (N sidechain)	
QS-13-5	**Hydrophobic contact**	
	Gly3—Tyr178	Gln7—Arg214
	**Hydrogen bond**	
		Gln7 (C=O)—Arg214 (N sidechain)

The nature of the interaction is classified as hydrophobic contact (the residues involved in the contact are specified) or hydrogen bonding (the residues and atoms involved in the contact are specified).

**Table 7 pharmaceuticals-18-00940-t007:** List of the interactions identified between the second most representative conformation of QS-13-derived peptides with a disulfide bridge and αvβ3 integrin.

Peptides with Disulfide Bond—Second Most Representative Conformation	αv	β3
QS-13-1	**Hydrophobic contact**	
	Lys11—Ala215	
	Ser13—Thr212	
	Ser13—Ala215	
	**Hydrogen bond**	
	Cter (N)—Thr212 (O sidechain)	
	Cter (N)—Ala213 (C=O)	
QS-13-2	**Hydrophobic contact**	
		Nter—Tyr122
		Gln1—Ser123
		Lys2—Ser123
		Ser13—Asp126
		Ser13—Met335
		Tyr12—Asp126
		Tyr12—Met335
		Cter—Asp336
QS-13-3	**Hydrophobic contact**	
	Nter—Tyr178	
	Nter—Ala215	
	Gln1—Ala215	
	**Hydrogen bond**	
	Gln1 (N)—Asp218 (O sidechain)	
QS-13-4	**Hydrophobic contact**	
		Gly3—Met335
		Ser4—Met180
		Ser4—Asn313
		Ser4—Met335
		Gly8—Met180
		Cys9—Tyr122
		Cys9—Ser123
		Gly10—Arg214
		Lys11—Asn215
		Tyr12—Ala218
	**Hydrogen bond**	
		Gly10 (C=O)—Asn215 (N)
QS-13-5	**Hydrophobic contact**	
	Gly8—Tyr178	
	Cys9—Tyr178	

The nature of the interaction is classified as hydrophobic contact (the residues involved in the contact are specified) or hydrogen bonding (the residues and atoms involved in the contact are specified).

**Table 8 pharmaceuticals-18-00940-t008:** List of the interactions identified between the most representative conformation of QS-13-derived peptides without a disulfide bridge and α5β1 integrin.

Peptides Without Disulfide Bond—Most Representative Conformation	α5	β1
QS-13-1	**Hydrophobic contact**	
	Ser3—Phe187	Arg5—Gln191
		Val8—Ser134
	**Hydrogen bond**	
		Ser4 (C=O)—Ser134 (N)
		Val10 (C=O)—Lys182 (N sidechain)
		Val10 (C=O)—Gln191 (N sidechain)
QS-13-2	**Hydrophobic contact**	
	Gly6—Trp157	Cys9—Tyr133
	Gln7—Phe187	
	**Hydrogen bond**	
		Cys9 (C=O)—Lys182 (N sidechain)
		Ser13 (C=O)—Ser132 (O sidechain)
		Ser13 (C=O)—Tyr133 (N)
		Ser13 (C=O)—Ser134 (N)
		Ser13 (C=O)—Ser134 (O sidechain)
QS-13-3	**Hydrophobic contact**	
	Cys9—Trp157	
	Val10—Trp157	
	Lys11—Trp157	
	Lys11—Ala159	
	**Hydrogen bond**	
		Gln7 (C=O)—Tyr133 (N)
QS-13-4	**Hydrophobic contact**	
	Nter—Phe187	Gly3—Ser134
	Nter—Asp227	Gly3—Ser227
	**Hydrogen bond**	
		Nter (C=O)—Ser227 (O sidechain)
		Gly3—(C=O)—Ser134 (O sidechain)
QS-13-5	**Hydrophobic contact**	
	Ser4—Asp227	Cter—Asn224
	Arg5—Ale225	Cter—Ser227
	Lys11—Phe187	
	**Hydrogen bond**	
	Cys6 (C=O)—Ser224 (O sidechain)	Cter (N)—Asn224 (C=O)
		Cter (N)—Asn224 (N)
		Gly3 (C=O)—Ser227 (O sidechain)
		Ser13 (N)—Asn224 (C=O)

The nature of the interaction is classified as hydrophobic contact (the residues involved in the contact are specified) or hydrogen bonding (the residues and atoms involved in the contact are specified).

**Table 9 pharmaceuticals-18-00940-t009:** List of the interactions identified between the second most representative conformation of QS-13-derived peptides without a disulfide bridge and α5β1 integrin.

Peptides Without Disulfide Bond—Second Most Representative Conformation	α5	β1
QS-13-1	**Hydrophobic contact**	
	Nter—Gly255	
	Nter—Asn256	
	Nter—Leu257	
	Gln1—Ile225	
	Gln1—Leu257	
QS-13-2	**Hydrophobic contact**	
	Nter—Trp157	Nter—Ser177
		Gln1—Pro186
		Gln1—Ser177
	**Hydrogen bond**	
	Lys2 (C=O)—Trp157 (N sidechain)	Nter (C=O)—Ser177 (N)
		Gln1 (N)—Pro186 (C=O)
QS-13-3	**Hydrophobic contact**	
	Lys11—Phe187	
	**Hydrogen bond**	
		Ser4 (C=O)—Gln191 (N sidechain)
		Ser13 (C=O)—Tyr133 (N)
QS-13-4	**Hydrophobic contact**	
	Nter—Ile225	
	Gln1—Asp227	
	**Hydrogen bond**	
	Gln1 (N)—Ser224 (C=O)	
QS-13-5	**Hydrophobic contact**	
	Cys6—Trp157	Gln7—Leu225
	Gly8—Trp157	Ser10—Leu225
	**Hydrogen bond**	
		Gln7 (C=O)—Lys182 (N sidechain)
		Ser10 (N)—Leu225 (C=O)

The nature of the interaction is classified as hydrophobic contact (the residues involved in the contact are specified) or hydrogen bonding (the residues and atoms involved in the contact are specified).

**Table 10 pharmaceuticals-18-00940-t010:** List of the interactions identified between the most representative conformation of QS-13-derived peptides with a disulfide bridge and α5β1 integrin.

Peptides with Disulfide Bond—Most Representative Conformation	α5	β1
QS-13-1	**Hydrophobic contact**	
	Tyr12—Phe187	Ser4—Tyr133
	**Hydrogen bond**	
		Ser4 (N)—Tyr133 (C=O)
QS-13-2	**Hydrophobic contact**	
	Nter—Phe187	
	Gln1—Phe187	
	Lys2—Leu128	
	Gly10—Phe187	
	Lys 11—Phe187	
	Gln12—Trp157	
	**Hydrogen bond**	
	Cys9 (C=O)—Trp157 (N sidechain)	Gln1 (C=O)—Lys182 (N sidechain)
		Ser4 (C=O)—Lys182 (N sidechain)
		Ser13 (C=O)—Ser132 (O sidechain)
		Ser13 (C=O)—Ser134 (O sidechain)
QS-13-3	**Hydrophobic contact**	
	Ser4—Phe187	Nter—Ser134
	Arg5—Phe187	Nter—Ser227
	—Asp227	
	**Hydrogen bond**	
	Nter (C=O)—Ser224 (O sidechain)	Gly3 (C=O)—Ser227 (O sidechain)
		Gln7 (C=O)—Ser134 (N)
QS-13-4	**Hydrophobic contact**	
	Nter—Phe187	Nter—Glu320
	Cys6—Ser224	
	Cys6—Ile225	
	Cys9—Ser224	
	Cys9—Ile225	
	Cys9—Asp227	
	**Hydrogen bond**	
		Nter (C=O)—Ser227 (O sidechain)
		Gly8 (N)—Glu320 (O sidechain)
		Lys11 (N))—Ser227 (O sidechain)
QS-13-5	**Hydrophobic contact**	
	Nter—Phe187	
	**Hydrogen bond**	
		Ser10 (C=O)—Ser227 (O sidechain)
		Lys11 (C=O)—Ser134 (O sidechain)

The nature of the interaction is classified as hydrophobic contact (the residues involved in the contact are specified) or hydrogen bonding (the residues and atoms involved in the contact are specified).

**Table 11 pharmaceuticals-18-00940-t011:** List of the interactions identified between the second most representative conformation of QS-13-derived peptides with a disulfide bridge and α5β1 integrin.

Peptides with Disulfide Bond—Second Most Representative Conformation	α5	β1
QS-13-1	**Hydrophobic contact**	
	Cys6—Ala159	Gln1—Asn224
	Val 8—Asp227	
	Cys9—Phe187	
	Cys9—Asp227	
	Tyr12—Ser224	
	**Hydrogen bond**	
	Lys11 (C=O)—Ser224 (O sidechain)	Gln1 (N)—Asn224 (C=O)
QS-13-2	**Hydrophobic contact**	
	Cys6—Phe187	
	Gly10—Ser224	
	Lys11—Ser224	
	Tyr12—Ser224	
	Ser13—Ser224	
	**Hydrogen bond**	
	Cter (N)—Ile (C=O)	
	Arg5 (C=O)—Trp157 (N sidechain)	
QS-13-3	**Hydrophobic contact**	
	Lys2—Trp157	Gln7—Lys182
	Lys2—Asp227	
	Gly3—Phe187	
	Gly3—Ser224	
	Gly3—Asp227	
	Gly8—trp157	
	Cys9—Trp157	
	Cter—Phe187	
	Cter—Ser224	
	**Hydrogen bond**	
	Ser13 (C=O)—Gln221 (N sidechain)	Gln7 (C=O)—Lys182 ( N sidechain)
QS-13-4	**Hydrophobic contact**	
	Tyr12—Phe187	
	**Hydrogen bond**	
		Gln7 (C=O)—Gln191 (N sidechain)
		Gly8 (C=O)—Tyr133 (N)
		Cys9 (C=O)—Ser134 (O sidechain)
QS-13-5	**Hydrophobic contact**	
	Gly8—Ala159	Ser13—Lys182
	Ser10—Phe187	
	Lys11—Phe187	
	Tyr12—Phe187	
	Ser13—Trp157	
	**Hydrogen bond**	
		Ser13 (C=O)—Lys182 (N sidechain)

The nature of the interaction is classified as hydrophobic contact (the residues involved in the contact are specified) or hydrogen bonding (the residues and atoms involved in the contact are specified).

## Data Availability

The original contributions presented in this study are included in the article/Appendix A. Further inquiries can be directed to the corresponding authors.

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
