# Peer review of "In Silico Prediction of Tetrastatin-Derived Peptide Interactions with αvβ3 and α5β1 Integrins"

_pharmaceuticals, 2025, doi:10.3390/ph18070940_

Round 1
Reviewer 1 Report
Comments and Suggestions for Authors
Dear Editor and Authors,
The manuscript titled “In silico prediction of Tetrastatin-derived peptide interactions with αvβ3 and α5β1 Integrins” is overall well written and addresses an important topic in the field of computational biology and peptide-integrin interactions. However, I believe the manuscript would benefit significantly from a major revision. The following points are suggested for your kind consideration:
- Keywords: Please arrange the keywords in alphabetical order to maintain consistency and adhere to standard academic conventions.
- Figures: The resolution of all figures needs to be improved to enhance clarity and readability, particularly for the structural representations.
- Literature Review: It is recommended to cite some recent publications (from the past 3–5 years) in the relevant area.
- Interaction Details: Kindly include π interactions (if any) in the interaction tables. Additionally, specify the atoms involved in hydrogen bonds and hydrophobic interactions.
- Docking Analysis: Please mention the binding energies of at least three docked poses in tabular form, along with the corresponding figures and explanations.
- Diffraction Studies: Include diffraction studies (X-ray or electron-based).
- Peptide Synthesis and Characterization: It would greatly improve the manuscript if the synthesis and characterization of the most potent peptide, as identified by docking and simulation studies, is included. This could be supported by relevant spectroscopic or analytical data (e.g., NMR, MS, HPLC), making the work more comprehensive and impactful.
I hope the above suggestions are helpful in enhancing the quality and presentation of the manuscript. Looking forward to a revised version that addresses these key aspects.
Comments on the Quality of English LanguageNeeds to be improved in more simple way to readers.
Reviewer 2 Report
Comments and Suggestions for Authors
This manuscript presents an in silico investigation of QS-13-derived peptides designed to improve aqueous solubility, assessing their potential to bind αvβ3 and α5β1 integrins at the RGD-binding interface. The study builds logically upon previous work on the native QS-13 peptide. The authors combine molecular dynamics simulations of the isolated peptides with molecular docking against ligand-free integrin static structures, offering insight into the structural determinants of integrin binding and potential anti-angiogenic properties.
The methodology is overall well described. However, the rationale for comparing peptides with and without disulfide bridges could be more clearly explained in the results section. For example, is there experimental evidence confirming whether disulfide bridges form under physiological conditions?
In addition, some important points need further development. First, only one ligand-free structure was used for each integrin in the docking studies. Given that integrins are flexible and may adopt different conformations upon ligand binding, it would be valuable to also consider ligand-bound structures and to perform MD simulations of the ligand-free proteins and the complexes and extract different conformations by clustering to account for protein flexibility. Second, maybe the best docking poses should be assessed by MD to assess the stability of binding and induced fit effects. This would provide a more robust representation of the binding landscape and could yield different energetic and structural outcomes.
Furthermore, I have some minor comments:
In section Results 2.1, Table I is incorrectly referenced (it should be Table 2), which lists the peptide sequences. Another issue concerns the inconsistent table numbering throughout the manuscript (switching between numeric and Roman numbers), which should be standardized. Additionally, while the comparison between docking poses and co-crystallized RGD peptides is an important validation step, the PDB IDs used for these comparisons should be explicitly stated in the main text (and the supplementary figure called) to support traceability.
Overall, addressing these points would significantly improve the scientific robustness and clarity of the manuscript.
Round 2
Reviewer 1 Report
Comments and Suggestions for Authors
The manuscript can be accepted for publication in its current form.